# Detection of Leachable Components from Conventional and Dental Bulk-Fill Resin Composites (High and Low Viscosity) Using Liquid Chromatography-Tandem Mass Spectrometry (LC-MS/MS) Method

**DOI:** 10.3390/polym15030627

**Published:** 2023-01-25

**Authors:** Matea Lapaš Barišić, Hrvoje Sarajlija, Eva Klarić, Alena Knežević, Ivan Sabol, Vlatko Pandurić

**Affiliations:** 1Private Dental Office, 10000 Zagreb, Croatia; 2Forensic Science Center, Ivan Vucetic, 10000 Zagreb, Croatia; 3Department of Endodontics and Restorative Dentistry, School of Dental Medicine University of Zagreb, 10000 Zagreb, Croatia; 4Division of Restorative Sciences, Herman Ostrow School of Dentistry, University of Southern California, Los Angeles, CA 90089, USA; 5Division of Molecular Medicine, Ruđer Boskovic Institute, 10000 Zagreb, Croatia

**Keywords:** liquid chromatography coupled with triple quadrupole tandem mass spectrometry (LC-MS/MS), dental composites, bulk-fill composites, elution, residual monomer

## Abstract

The aim of this study was to investigate leachable components (monomers) in high and low viscosity dental bulk-fill resin composites and conventional resin composite materials after polymerization. Six bulk-fill and six conventional dental resin composite materials were used in this study. The samples of each material (three sets of triplicates) were cured for 20 s with irradiance of 1200 mW/cm^2^ with a LED curing unit and immersed in a 75% ethanol solution at 37 °C. The eluates from each triplicate set were analyzed after 24 h, 7 days or 28 days using liquid chromatography coupled with triple quadrupole tandem mass spectrometry (LC-MS/MS). Detectable amounts of 2-Hydroxyethyl methacrylate (HEMA) were found in both Gradia materials and the amount observed across different time points was statistically different (*p* ˂ 0.05), with the amount in solution increasing for Gradia and decreasing for Gradia Direct flo. Bisphenol A diglycidildimethacrylate (BIS GMA) was found in Filtek and Tetric materials. Triethylene glycol dimethacrylate (TEGDMA) was detected in all materials. On the other hand, there were no statistically significant differences in the amounts of TEGDMA detected across different time points in either of the tested materials. Monomers HEMA, TEGDMA, 4-dimethylaminobenzoic acid ethyl ester (DMA BEE) and BIS GMA in bulk-fill and conventional composites (high and low viscosity) can be eluted after polymerization. The good selection of composite material and proper handling, the following of the manufacturer’s instructions for polymerization and the use of finishing and polishing procedures may reduce the elution of the unpolymerized monomers> responsible for the possible allergic and genotoxic potential of dental resin composites.

## 1. Introduction

Light-cured dental composites are the materials of choice in restorative dentistry due to their esthetic properties, mechanical strength and applicability in minimally invasive procedures. The time-consuming incremental technique has recently been replaced by a bulk technique thanks to the discovery of bulk-filling composites. Recently, a brand-new class of resin-based composites known as bulk-fill composites has been introduced. Their main selling point is the ability to install and cure increments of up to 4 mm in a single step, reducing chairside time. These composites also have a fast activation time due to newly designed initiation mechanisms and higher translucency due to larger filler particles and lower filler loading. Bulk-fill composites simplify the clinical procedure because they can be used in thicker layers [1]. For this reason, manufacturers claim that the composite can control the polymerization process and ensure adequate depth of cure even when larger increments are used. The most important advantage offered by these materials is the time saved in placing the material and in polymerization, as well as the reduced sensitivity to the technique [2]. The molecular basis of these resin composites has been altered to allow greater incremental incorporation by reducing or replacing Bis-GMA, resulting in a lower viscosity monomer, and/or by replacing higher molecular weight monomers often based on Bis-EMA, TEGDMA, EBPDMA, and UDMA monomers. The incorporation of stress reducers and modification of filler content also contribute to the reduction of polymerization shrinkage. When bulk-fill composites are used, polymerization shrinkage should be reduced, which in turn allows for good marginal integrity and less cusp deformation in the final composite restoration [3].

The degree of conversion during polymerization refers to the ratio of monomer to polymer. In other words, the degree of conversion refers to the percentage of C=C bonds of the monomers present in the polymeric matrices that have undergone reaction. The internal standard refers to the percentage of C=C bonds determined from the ratio of cured to uncured monomers. The complete conversion of all monomers to polymers results in a conversion rate of 100 percent, but this is never achieved. The conversion rate is normally between 43 and 70%. Ten percent of the elution from resin composites is caused by free monomers [4]. Lower conversion rates result in more monomer eluting into the oral environment, which negatively affects the mechanical and physical properties of the material. Dental composites essentially consist of glass filler particles dispersed in methacrylate resin. The latter is photocurable and can be cured by radical polymerization when irradiated with visible light. The polymerization of multifunctional methacrylate monomers results in a densely crosslinked network and yields monomer conversions that rarely exceed 80% [5]. This suggests that residual monomers can elute from the restoration to the oral cavity. Dental composites consist of a few main components: organic matrix (monomers: 2 Hydroxyethyl metacrylate (HEMA), Bysphenil-glycidyl-methacrylate (Bis GMA) and/or Urethane-dimethacrylate (UDMA)), co-monomers (Ethylene glycol dimethylacrylate (EGDMA), Methyl ether methacrylate (DEGMA), Triethylene glycoldymethacrylate (TEGDMA)), inorganic fillers (quartz, borosilicate, lithium aluminum silicate glasses and amorphous silicas), photoinitiators (camphorquinone CQ, Lucirin TPO, PPD), co-initiator Ethyl 4-dymethyloamino benzoate (DMA BEE), inhibitors of polymerization (BHT) and photostabilizers (Benzophenone) [6,7,8]. Unreacted monomers might have an influence on the biocompatibility of the restorations and can cause local or systemic toxic effects [9,10,11,12]. The majority of the degradation products have probably not yet been identified. Lower conversion rates result in more monomer eluting into the oral environment, which negatively affects the mechanical and physical properties of the material. In water, 2–6 wt% in 70% ethanol, and 10% in methanol, the amount of eluting molecules varies [13]. The aging of composites can also lead to more porosity due to filler wear, water sorption, and chemical/enzymatic degradation, resulting in an increased release of unpolymerized monomers originally trapped in the polymer network. The ability of the residual monomers to penetrate the matrices and expand the space between the polymer chains, allowing the soluble chemicals to diffuse, was the reason for the use of ethanol as a solvent in the present investigation. It is claimed that the replacement of the composite in the oral cavity releases various components. Reportedly, these substances have estrogenic, genotoxic, mutagenic, and cytotoxic properties. Reportedly, the unpolymerized monomer can reach the pulp and cause negative pulpal reactions [14]. In order to reduce polymerization shrinkage, achieve adequate depth of cure and reduce the elution of components from conventional composites, it is necessary to apply the material in layers, whereas bulk-fill dental resin composites use the single-layer technique to achieve the same [15]. There are few literature data on the elution of monomers from bulk-fill composites. Polydorou et al. [16] investigated the elution of monomers from two conventional dental composites after different polymerization and storage times using LC-MS/MS. No significant difference was found between samples polymerized for 20 and 40 s, and only BisGMA and TEGDMA were detected. Manojlovic et al. [17] quantified the elution of the major monomers from four commercial composites using high-performance liquid chromatography and established a mathematical model of the elution kinetics. It was shown that TEGDMA was identified as the main compound released from dental composites analyzed by high-performance liquid chromatography (HPLC) [18]. Mass spectroscopy can basically be conceptualized as molecular scale. Tandem mass spectrometers (LC-MS/MS), sometimes referred to as MS/MS instruments, are devices used to chemically process molecules before weighing the results. Mass spectrometers use charged molecules (ions) in a vacuum to make these observations. HPLC, on the other hand, works with molecules in solution. The first step at MS is to convert the sample into a charged ion in the gas phase, which is followed by the measurement. While liquid chromatography separates mixtures with multiple components, mass spectrometry provides spectral information that can help identify (or confirm the suspected identity of) the individual separated components. It is a highly efficient chemical technique that combines the physical separation capabilities of liquid chromatography with the mass analysis capabilities of mass spectrometry. HPLC is less accurate and sensitive than LCMS, which was advantageous for this study to more accurately determine the elution of the monomers of the most commonly used high and low viscosity composites for dental fillings [19].

The objective of this study was to determine all possible residual monomers from conventional and bulk-fill composites (high and low viscosity) leaching from the materials after different time intervals (24 h, 7 days, and 28 days) using more accurate and sensitive liquid chromatography coupled with triple quadrupole tandem mass spectrometry (LC-MS/MS). For this study, the following two working hypotheses were made: (1) there is no elution of residual monomers after polymerization of conventional and filled resin composites; (2) there is no difference in elution between conventional and filled composites with high and low viscosity; (3) there is no difference in monomer elution measured after 24 h, 7 days and 28 days.

## 2. Materials and Methods

### 2.1. Sample Preparation

In this study, six commercially available bulk-fill dental composites were investigated and compared with six conventional composites (Table 1). Low- and high-viscosity composite samples were prepared by applying the resin material directly from the compule into a Teflon mold (diameter 5 mm and depth 2 mm). After placing the composite resin, the surface was covered with a transparent plastic Mylar strip, and the sample was light-cured according to the standard protocol (20 s of irradiation with 1200 mW/cm^2^ in a wavelength range of 380–515 nm) using a LED-curing device, Bluephase G2 (Ivoclar Vivadent, Schaan, Liechtenstein), which was measured with the LED light-curing radiometer Bluephase Meter II (Ivoclar Vicadent, Schaan, Liechtenstein) and immersed in 5 mL of 75% ethanol solution at 37 °C. After a dark storage period of 24 h, 7 days or 28 days, the eluates were collected and analyzed (Figure 1). The study was approved by the Ethics Committee of the Faculty of Dentistry, College of Zagreb (number 05-PA-30-XXI-10/2020). For the measurement of the monomer elution and due to the high effectiveness of the LC-MS/MS method, a sample size of 5 samples was determined to be optimal for the study. The sample size was calculated using the G * power program based on the difference in numerical variables between measurements, setting a significance level of 0.05 and a power level of 0.8 (high power size of 0.8), and obtaining a minimum required sample size of 5 samples (number of replicates within each experimental group) per group.

### 2.2. Analytical Technique

Liquid chromatography coupled with triple quadrupole tandem mass spectrometry (LC-MS/MS) was used to evaluate the presence of leachable compounds in the eluates. Sample preparation was performed as follows: 0.5 mL of ethanol-water extract was concentrated in vacuo (Martin Christ, Osterode am Harz, Germany) until dry, and the sample was reconstituted in 100 μL methanol. Quantitative and qualitative analysis was reconstituted using LC-MS/MS (Shimadzu LC system AC 20 coupled to ABSciex 3200 Qtrap tandem mass spectrometer system). All compounds were determined qualitatively by comparing their production mass spectra to the available internal production mass spectra library, and TEGD-MA, Bis GMA, HEMA, CQ and DMABEE were also measured quantitatively using calibration curves. Calibration curves were constructed with exactly 6 concentration levels (Eurachem Guide) ranging from 100 ng/mL–10,000 ng/mL (CQ and DMABEE), 1000 ng/mL–50,000 ng/mL (TEGDMA and Bis GMA), and 100 ng/mL–20,000 ng/mL (HEMA). Components were separated on a Poroshell 120 EC -C18HPLC column 2.1 × 100 mm 2.7 µm (Agilent Technologies), and 5 µL of sample was added via the autosampler. The column was maintained at 40 degrees Celsius. The mobile phases used were 63 mg of ammonium formate and 1 mL of formic acid to 1 L of deionized water (mobile phase A) and LiChrosolv methanol (mobile phase B). Separation was performed using a constant total flow rate of 400 µm/min, 30% of mobile phase B. A gradient flow was introduced after 0.5 min, reaching 100% of B after 3 min, held isocratic until 3.75 min and then decreased to 30% B at 3.76 min and held isocratic until the end of the run at 6.5 min. The mass spectrum was set to MRM mode for quantification using Solvent Blue 35 (SB -35) as the internal standard.The relationship between concentration and absorbance was plotted using the calculated areas under the peaks. The percentages of the different polymers in each study group were calculated. All measurements were performed once for each sample. The measurements were performed after 1, 7, and 28 days, respectively [20].

### 2.3. Statistical Test Methods

Data were recorded in an Excel spreadsheet (Microsoft, Seattle, WA, USA), and the mean of the technical replicates for each material and time point was calculated. Results were expressed as mean and standard deviation (SD). Statistical analysis and plots were generated using Medcalc (v11.4, MedCalc Software bv, Ostend, Belgium). Normality of the data was assessed with the Kolmogorov-Smirnov test. The amount of each monomer released at different time points was assessed using one-way analysis of variance or the Kruskal-Wallis test (one-way analysis of variance by ranks) and repeated-measures nonparametric Friedma analysis of variance. Comparisons between specific groups of materials at the same time point were made using ANOVA. In addition, individual comparisons between 2 groups were assessed with the t test. A *p* value less than 0.05 was considered statistically significant.

## 3. Results

The overall results showed that monomers leached from the polymerized samples into the ethanol were detected at most time points. The mean amounts (and standard deviation) of each monomer leached from each material at each time point are shown in Table 2. DMA-BEE was found in all samples analyzed, but the concentration detected in Gradia and Gradia Direct flo was well below the limit of quantitation (LOQ) of the method and was therefore considered to be zero. A statistically significant difference was found between the different time points in some of the materials analyzed (Fbf *p* = 0.006, FBFf *p* = 0.002, SDR *p* = 0.004 and TEC *p* = 0.001 samples, respectively). In each case, there was an increase in the concentration of the leachable component in the solution. *Bis GMA* was detected in nine different materials (Table 2). There was a statistically significant difference between time points in TeCBf (*p* = 0.001), FBFf (*p* = 0.002), TEC (*p* = 0.033), Fsf and FBf (*p* < 0.001) samples. The amount of leached compounds increased in samples TeCBf, FbFf and TEC and decreased in samples FSf and FBf with time. Detectable amounts of HEMA were found in only two of twelve sample materials. In both Gradia and Gradia Direct flo, the amount detected at different time points was statistically different, increasing in solution in Gradia (from 14.1 + 1.6 to 34.2 + 10; ANOVA *p* = 0.022) and decreasing in Gradia Direct flo (from 9.7 + 1.1 to 7.7 + 0–5; ANOVA *p* = 0.036). TEGDMA was detected in all materials. On the other hand, there were no statistically significant differences in the amounts detected in any of the tested materials at the different time points (Table 2). To evaluate the difference between conventional and bulk-fill materials or between low- and high-viscosity materials, the eluted values at the earliest time point for each analyte were compared between the different types. To reduce variability due to different commercial suppliers, results from the same supplier were grouped where possible (Table 3). For filtek materials, there was a statistically significant difference in the elution of DMA BEE between bulk (Filtek Bulk fill) and conventional (Filtek Supreme) high viscosity materials (BHV vs. CHV; *t*-test *p* value = 0.006). The difference between bulk fill (Filtek Bulk Fill flow) and conventional (Filtek Supreme flow) low viscosity composites approached but did not reach the significance threshold of 0.05 (BLV vs. CLV; *t*-test *p* = 0.056). There was no significant difference between high- or low-viscosity materials. Interestingly, the significant differences initially observed were not significant after the compounds were leached for 7 days or after 28 days (Table 4; Figure 2). The only comparison that reached statistical significance was the difference between DMA BEE, which was leached from conventional high- and low-viscosity filtek materials. The difference was barely significant and was mainly due to a lower variance in the measurements (*t*-test *p*-value *p* = 0.049; Table 4). There were statistically significant differences in all comparisons between TetricEvo Bulk and conventional materials and between high- and low-viscosity composite materials for DMA BEE (all *p*-values < 0.01). However, for the other composite materials (Gradia, Gradia Direct flo, X-tra fil and SDR), there was only a statistically significant difference between bulk-fill high and bulk-fill low viscosity (*p* = 0.006); other comparisons were not possible. After 28 days, most of the differences between the leached compounds diminished, and only the difference between DMA BEE, which was leached from conventional high- and low-viscosity composites, remained significant (*t*-test *p* = 0.022; Table 4; Figure 3). *Bis GMA* was not detected in the low-viscosity Filtek Bulk material, so some comparisons were not possible. In other cases, there was also no statistical significance of the leached BiS-GMA amounts in the other Filtek materials after 24 h (Table 3). At the final time point after 28 days, Bis GMA was not observed in the low-viscosity bulk, again preventing some comparisons. The remaining comparisons were not statistically significant. As with DMA BEE, the amounts of *BiS GMA* were significantly different in all Tetric materials (all *p* < 0.01) and were not detected at all in the other materials tested after 24 h. After 28 days, as with DMA BEE, the differences between materials were less significant and only the high-viscosity bulk materials and the conventional high-viscosity materials remained statistically significantly different (*p* = 0.001). *Bis GMA* was also not observed in the other materials after 28 days (Table 4). The differences in leached TEGDMA were significant only between high- and low-viscosity filtek materials (*p* = 0.037). On the other hand, the TEGDMA differences were significant only when comparing bulk-fill and conventional Tetric materials. The comparisons of the other materials were not statistically significant after 24 h, but after 28 days with TEGDMA the differences were more significant for the Filtek materials (Table 4). No significant differences were observed in Tetric and other materials. Since HEMA was not detected in any of the Filtek or Tetric materials, no comparisons could be made. The amounts observed in Gradia and Gradia Direct flo (high- and low-viscosity versions of the same material) were statistically significantly different at both 24 h (*p* = 0.018) and 28 days (*p* = 0.045). The large differences between the high- and low-viscosity materials are shown in Figure 3.

## 4. Discussion

The aim of various previous studies was to determine the constituents extractable from polymerized resin composites. In most studies, only a few substances could be identified [21,22]. An important factor affecting the leaching of monomers is the type and molecular size of the monomers in the resin. Smaller molecules are leached faster than larger ones, and monomers with small molecular weight can be extracted in larger amounts than monomers with large molecular weight [23]. The various analytical methods used to determine leachable species from resin composites have been described by Ruyter and Oysaed [24]. In this study, we used liquid chromatography-tandem mass spectrometry (LC-MS/MS) to identify and quantify the elution of monomers. With the exception of Polydorou et al. [16], who studied the elution of monomers from two light-cured materials (nanohybrid and ormocer) after different curing times and different storage times, there is not much literature on the release of monomers from composites using this method. It is well known that eluted monomers can contribute to the cytotoxicity of composite resins. Geurtsen and Leyhausen [25] reported that cytotoxic aqueous resin eluates often contain high amounts of TEGDMA. In fact, the National Institute of Occupational Safety and Health has classified TEGDMA as an irritant to various tissues [26,27,28]. Spahl et al. [29] showed in their study that co-monomers and various additives, as well as impurities from the manufacturing process, were detected in all polymerized resin composites. Several in vitro studies have shown cytotoxic, genotoxic, mutagenic or estrogenic effects on the pulpal and gingival/oral mucosa due to the reactions of some monomers [30,31].

In the present study, the elution of TEGDMA, Bis GMA, HEMA, and DMA BEE from conventional and bulk-fill resin composites was investigated at three time intervals. The first and second hypotheses were rejected because residual monomers eluted after the polymerization of the materials, and there were also differences in elution between conventional and bulk-fill resin composites. The results of this study showed that TEGDMA was detected in all the materials studied, but there were no statistically significant differences in the amounts detected at different time points in any of the materials studied. This is consistent with other studies that have also found TEGDMA to be the main monomer eluting from composite resins [25]. However, differences in leached TEGDMA were significant only between high- and low-viscosity Filtek materials and bulk and conventional Tetric materials, and only after 28 days, with TEGDMA differences being more pronounced in Filtek materials. TEGDMA is a small monomer and elutes faster than larger molecules such as Bis GMA [32]. In this study, Bis GMA was detected in nine different materials. Only SDR, Gradia, and Gradia Direct flo did not contain Bis GMA at any of the time points. SDR is a flowable, single-component, fluoride-containing, visible-light-cured, radiopaque posterior composite restorative material designed for use as a base for Class I and II preparations. It has the typical handling characteristics of a “flowable” composite, but can be used in 4 mm increments with minimal polymerization stress. According to the manufacturer, eluted monomers from SDR can also irritate the skin, eyes, and oral mucosa [33]. HEMA is used in dental composites due to its hydrophilic application as a co-monomer of the organic resin matrix and was found in only a few tested materials. HEMA is known to cause cytotoxic and genotoxic effects [34]. In Gradia and Gradia Direct flo, the amount was statistically different, increasing in Gradia and decreasing in Gradia Direct flo. HEMA could be a degradation product of UDMA, which is a component of Gradia and Gradia Direct flo according to the MSDSs. Bis GMA was not detected in the low-viscosity material, so some comparisons were not possible and there was no statistical significance of the leached Bis GMA amounts in filtek materials after 24 h and at the last time point after 28 days. Bis GMA was not observed in the low viscosity bulk material, again preventing some comparisons. The amounts observed in Gradia and Gradia Direct Flo were not statistically significant at both 24 h and 28 days. In the study by Cebe et al., [2] the amount of eluted Bis GMA from Tetric Evo Ceram Bulk Fill and the amount of eluted TEGDMA and HEMA from X-tra Fill were higher than other composites, which was in contrast to our study where the TEGDM monomer was more eluted from all types of high- and low-viscosity bulk composites compared to Bis GMA, while HEMA was found only in Gradia composites.

DMA BEE is a co-initiator used in composites to accelerate the degradation of initiators into radicals and thus polymerization [35]. Various solvents such as distilled water, saliva, ethanol, methanol and acetonitrile have been used in studies to evaluate the leaching of monomers [25]. A 75% ethanol/water solution was the solution of choice in several studies to simulate and accelerate the aging of restorations [26]. The oral cavity represents an environment somewhere between water and more aggressive solvents (ethanol, methanol, acetonitrile) [30]. A 75 percent ethanol/water solution has a solubility parameter very close to that of oral fluid, resulting in maximum softening of the resin [36,37]. This solution is recommended by the United States Food and Drug Administration (FDA) guidelines (1976, 1988) as a clinically relevant mouth-simulating fluid and has been used in several studies [38,39,40]. Therefore, this solution was used in this study. The elution time of 24 h is based on previous findings [28] suggesting that almost all leachable substances are eluted within 24 h after polymerization. However, the elution of monomers is definitely not linear over time, and there are studies showing that the release of monomers lasts up to 30 days [16,37] or even up to one year after polymerization [16]. Therefore, additional time points of 7 and 28 days were also investigated. In this study, a statistically significant difference in the release of DMA BEE was found between Filtek Bulk and conventional high-viscosity materials and the difference between Filtek Bulk and conventional low-viscosity composites. The only comparison where a statistically significant difference was obtained was the amount of DMA BEE leached from conventional high viscosity and conventional low viscosity Filtek materials, but the difference was hardly significant and was mainly due to a smaller variation in the measurements. In all comparisons for Tetric Bulk and conventional and high- and low-viscosity preparations, there were statistically significant differences for DMA BEE, and there was also a statistical difference between X-tra fill and SDR, but other comparisons were not possible. After 28 days, the leaching DMA BEE of the conventional high- and low-viscosity material remained significant. There were also statistically significant differences between time points for samples TCbf, FBFf, TEC, FSf and FBf, and the amount of compound leached increased for samples TECbf, FBFf, TEC and decreased for samples FSf and FBf over time.

HEMA release showed a maximum increase on the 28th day for Gradia, which was in accordance with Altıntaş and Üşümez, [41], who investigated the residual monomer release from resin cements and reported the HEMA release amount from Nexus 2 (Kerr/Italy) cement to be lower in the 10th minute and much higher on the 21st day. Gradia Direct flo showed maximum increase in the first 24 h, with decreasing amounts of leached monomer after 28 days. This was similar to a study by Duruk et al. [42], who found that the amount of HEMA released from the resin cement of Ionolux (VOCO, Cuxhaven, Germany) was found to be very low for the 1st hour and higher on the first day in comparison to the 21st day. This situation may be due to the interaction of HEMA molecules with water, considering that HEMA is highly hydrophilic and the solution consists of 75% ethanol–25% water. For TEGDMA, the circumstances were different among materials because in some materials, TEGMA was higher after only 24 h compared to 28 days (X-tra Fil, SDR, Filtek Supreme, TetricEvo Ceram and Gradia Direct flo), and the highest amounts were found after 28 days for Filtek Bulk Fill, Tetric Evo Ceram Bulk Fill, Tetric EvoFlow Bulk Fill, Gradia and TetricEvo flow. The differences in filler particle type and monomer ratios specified by the manufacturer are assumed to be the cause of the residual monomer release seen between micro-hybrid and nano-hybrid composites. After one day and fourteen days, De Angelis et al. [43] used HPLC to evaluate the eluted monomer from the GrandioSO (VOCO) nanohybrid composite. According to their findings, TEGDMA levels became detectable after 24 h, while BIS-GMA levels were higher after 24 h than after 14 days. According to Duruk et al., after 24 h and 14 days, the amounts of TEGDMA released by the GrandioSO composite were much higher than on the 21st day when it was undetectable [42]. Additionally, after 24 h the amounts of BisGMA were higher than after 7 days or 28 days for Filtek Bulk Fill, Filtek Supreme flow, and TetricEvo flow, while for all others, the composites’ amount of Bis GMA were higher after 28 days except for SDR, G, and GDf; no BisGMA were determined in either of the measured periods.

In the majority of studies, dilute ethanol, distilled water, and methanol have been used as solvents for testing the materials. In other protocols, elution was also studied in artificial saliva and various media commonly used for cell culture development. Artificial saliva and distilled water are both water-based solvents that can simulate intraoral conditions. Greater dissolution efficiency is a characteristic of organic solvents, likely due to better sorption, swelling, and penetration of the material. Since monomers are usually hydrophobic, similar differences between the main release in organic solvents and those based on water have been found in experiments [9]. In in vitro studies on dental materials and their properties, the environment of the oral cavity is usually mimicked to ensure the repeatability and stability of the applicable analytical procedures. Saliva is constantly produced in the oral cavity to clean the surfaces of teeth and dentures before being excreted by swallowing. Natural human saliva has a very complex and diverse composition that is influenced by numerous individual factors (including food intake, bacterial colonization, and others) that fundamentally affect intraoral pH. Because of these factors, it is difficult to produce a synthetic formula that exactly matches real saliva [44]. However, because real human saliva is unstable outside the oral canal, its use for this purpose is also unreliable. It appears to be very difficult to replicate the exact intraoral conditions, and this should be taken into account when evaluating the results of this or any other in vitro research that cannot fully correlate with the in vivo situation. Studies that looked at monomer solutions in artificial saliva also confirmed that the elution of bulk-fill composites was equivalent to that of conventional materials, despite their greater incremental thickness. The hydrophobicity of the base monomers and the final network properties of the resin matrix have a significant influence on monomer elution [45].

With higher monomer concentrations in the samples stored for 1 month compared to those stored for 24 h and 7 days, it was found in the present study that increasing the storage time resulted in higher amounts of Bis GMA and DMA BEE elution for all Tetric and Filtek composites, as noted in the study by Janani et al. [46]. Nazar at al. also used high-performance liquid chromatography analysis and reported that longer storage times resulted in statistically significant increases in BisGMA and UDMA amounts for both Tetric and Filtek materials [47]. There are few studies investigating the long-term elution of monomers over 1, 3, and 12 months using liquid chromatography tandem mass spectrometry and high-performance liquid chromatography [16,46]. However, the long-term effects of residual monomers on biocompatibility are still unclear. Due to the constant salivary flow in the oral environment, monomer concentrations are not expected to reach the cumulative levels determined in this study, while long-term chronic exposure and systemic adverse effects must also be considered when evaluating the potential toxicity of eluted compounds.

The monomer released from Gradia materials was HEMA (2-hydroxyethyl methacrylate). It is a tiny, low-molecular-weight monomer that is soluble in both types of solvents (130 g/mol). HEMA is a commonly used co-monomer in commercial resin-based products because its hydrophilic properties prevent the separation of water and hydrophobic co-monomers. However, some unfavorable physico-mechanical properties of HEMA have been documented, such as low conversion efficiency and water retention, which hinders effective polymerization [48]. In addition, HEMA showed some cytotoxicity that affected cell survival [49], which could be exacerbated by the water solubility of HEMA. The TEGDMA monomer was found in comparatively high amounts in all tested materials, especially in organic solvents. TEGDMA is a low-viscosity, low-molecular-weight molecule (286.32 g/mol) that is often added to composites to reduce the viscosity of the mixture and thus increase the degree of conversion (DC). Unfortunately, the larger DC of TEGDMA also leads to greater shrinkage of the material during polymerization. For this reason, TEGDMA is often replaced, at least in part, by another monomer that has a larger molecular mass and lower viscosity (e.g., Bis-EMA). There are reports of the cytotoxic effect of TEGDMA on human and gingival fibroblasts clinically associated with pulp infarction and necrosis [45]. As in other studies, our study confirmed that Bis-GMA has the lowest release, as it has the highest molecular mass (512.599 g/mol) and the lowest solubility in all types of solvents. Due to its high refractive index, low volatility, strong mechanical properties, low volumetric shrinkage after polymerization, diffusivity into tissue, and good adhesion to enamel, Bis GMA is a basic matrix compound that is generally useful [50]. However, the market for materials based on Bis GMA resins [51] such as composites based on Bis EFMA has begun to expand due to concerns about the viscosity of Bis GMA, which can negatively affect the mechanical properties of materials, and its potential cytotoxic effect in combination with BPA [52]. Bezgin et al. [53] measured the release of residual monomers with HPLC after 24, 48, and 72 h and also determine the effects of finishing and polishing procedures on the elution of Bis-GMA, TEGDMA, UDMA, and HEMA monomers from compomer and bulk-fill composite resins. The finishing and polishing procedures had a significant effect on reducing the quantity of UDMA release, so the Mylar strip also used in our study did not prevent the formation of the oxygen-inhibition layer, and final polishing was still essential to remove the resin-rich outer layer, which can be the source of unreacted monomers that elute into the oral cavity.

Chemicals are released in order of cytotoxic potential as determined by Reichl et al.: HEMA < TEGDMA ˂ UDMA ˂ Bis GMA. In their cytotoxicity study, they found that the EC50 values for HEMA and TEGDMA decreased from about 5 mmol/L (6 h) to about 0.6 mmol/L (48 h) and from about 3 mmol/L (6 h) to about 0.4 mmol/L (48 h), respectively. [54]. In this study, human gingival fibroblasts were exposed to Bis-GMA at a concentration of 0.087 mmol/L, UDMA at a concentration of 0.106 mmol/L, and HEMA at a concentration of 11.530 mmol/L. Such a decrease in the viability of TEGDMA was observed at 3.460 mmol/L. When dental resin materials with and without Bis-GMA were compared, those that released Bis-GMA and TEGDMA were found to have a higher potential for cytotoxicity and genotoxicity [48]. Numerous studies [55,56] have described the specific effects of monomers placed in direct contact with dental pulp cells, including inflammation and suppression of dentin mineralization. In our study, the TEGDMA concentrations of all tested materials were found to be below the hazardous concentrations for TEGDMA identified in some previous studies [25,43].

To determine the quantity of released compounds, most previously cited studies performed the analysis prevalently through the HPLC (high-performance liquid chromatography) or GC–MS (gas chromatography mass spectrometry) methods. The analytical methods of LC–MS (liquid chromatography mass spectrometry) and UPLC-MS/MS (ultraperformance liquid chromatography-tandem mass spectrometry) were used rarely and not so often, so we compared our results to other studies dealing with this method but also with HPLC, which is much popular in this type of study. LC–MS (liquid chromatography mass spectrometry) technique is based on the detection of the mass-over-charge ratio of a compound of interest and its daughter ions, leading to two extra parameters that are compound-specific. Susila et al. [57] measured the elution of the composites using Liquid Chromatography-Mass Spectrometry (LC-MS). They measured BisEMA, BisGMA, TEGDMA and UDMA elution in three different materials: polysiloxane-dimethacrylate (Ceram XTM), Silorane (Filtek P90TM) and dimethacrylate (RestofillTM). Dimethacrylate-based composites eluted more monomer and exerted strong cytotoxicity, which was similar to results found in our study where monomers from bulk-fill and conventional composites (high and low viscosity) were eluted from 2 mm thick samples after polymerization of 20 s with irradiance of 1200 mW/cm^2^ with a LED curing unit.

Our results show that there are significant differences in the leachable components, depending in part on the type and consistency of the dental material studied. This was a pilot study with a smaller number of samples, and no correction was made for multiple testing. The number of samples tested for each material is a limiting factor for the study, but due to the large number of materials tested (which accounts for the uniqueness of this study) at multiple time points, it was not possible to increase the number of replicates for each material. The results should be confirmed with a larger number of samples, which is planned for future studies.

## 5. Conclusions

Within the limitations of the present quantitative study, it can be concluded that monomers (HEMA, TEGDMA, DMABEE, Bis GMA) can be eluted in bulk fill and conventional composites (high and low viscosity) after polymerization. The results indicate that the effect may be ambiguous, as apparently materials from different manufacturers release some monomers more than others. However, all but one material showed a high release of TEGDMA. The results of the present study show that the restorative materials investigated here are not chemically stable after polymerization, and the concentrations of eluted monomers can reach critical toxicity levels even after a single 2 mm thick restoration placement. Also, Mylar strips do not prevent the formation of the oxygen inhibition layer, and final polishing is still essential for the removal of the resin-rich outer layer, which may be the source of unreacted monomers eluting into the oral cavity. Thus, a good selection of composite material and proper handling, the following of the manufacturer’s instructions for polymerization, and the use of finishing and polishing procedures can reduce the release of unpolymerized monomers from composite materials with possible genotoxic and cytotoxic potential to soft tissues and to the body in general.

## Figures and Tables

**Figure 1 polymers-15-00627-f001:**
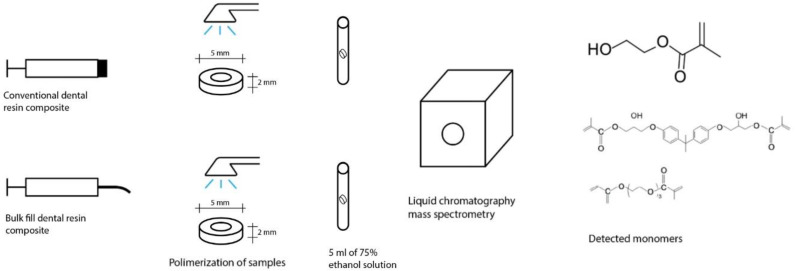
Sample preparation and monomer detection.

**Figure 2 polymers-15-00627-f002:**
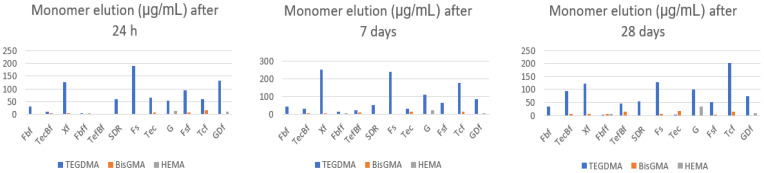
Mean amounts of leached TEGDMA, Bis GMA and HEMA at different time point. Fbf (Filtek Bulk Fill) TecBf (Tetric Evo Ceram Bulk Fill) Xf (X-tra Fil) Fbff (Filtek Bulk Fill flow) TefBf (Tetric EvoFlow Bulk Fill) SDR (SDR) Fs (Filtek Supreme) TeC (TetricEvo Ceram) G (Gradia) Fsf (Filtek Supreme flow) Tcf (TetricEvo flow) GDf (Gradia Direct flo). TEGDMA (Triethylene glycol dimethacrylate) Bis GMA (Bysphenil-glycidyl-methacrylate) HEMA (2-Hydroxyethyl methacrylate).

**Figure 3 polymers-15-00627-f003:**
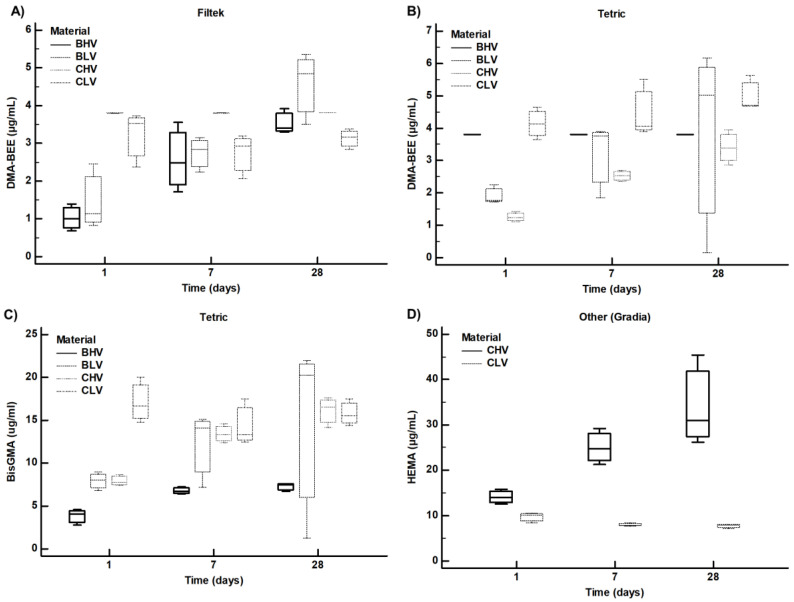
Amounts of leachable compounds detected by LC-MS/MS across different dental materials from the same provider (Filtek Panel (**A**); Tertic Panels (**B**,**C**); Gradia Panel (**D**)) in ethanol medium incubated at 37 Degrees Celsius for 24 h, 7 days or 28 days. BHV—Bulk high viscosity material, BLV—Bulk low viscosity material, CHV—Conventional high viscosity, CLV—Conventional low viscosity. Each material was sampled in three replicates and each replicate is represented by a mean of three LC-MS/MS measurements. Bis GMA (Bysphenil-glycidyl-methacrylate), DMA BEE (4-dimethylaminobenzoic acid ethyl ester), HEMA (2-Hydroxyethyl methacrylate).

**Table 1 polymers-15-00627-t001:** Composite materials used in study.

Name	Producer	Lot	Abbreviation	Matrix Composition Declared by Producer
Tetric Evo Ceram Bulk Fill	Ivoclar Vivadent, (Schaan, Lichtenstein)	82 O135539	TeCBf	Bis GMA, Bis-EMA, UDMA
Tetric EvoFlow Bulk Fill	Ivoclar Vivadent, (Schaan, Lichtenstein)	U34907	TefBf	Bis GMA, Bis EMA, UDMA
X-tra Fil	VOCO (Cuxhaven, Germany)	1438592l	Xf	Bis GMA, TEGDMA, UDMA
Filtek Bulk Fill	3M ESPE (St. Paul, MN, USA)	N626709	Fbf	Bis GMA, Bis EMA, UDMA, Procrylat resin
Filtek Bulk Fill flow	3M ESPE (St. Paul, MN, USA)	N732765	Fbff	Bis GMA, Bis EMA, UDMA, Procrylat resin
SDR	DENTSPLY (Charlotte, NC, USA)	1610131	SDR	Modified UDMA, EBPADMA, TEGDMA
Gradia	GC (Tokyo, Japan)	1710312	G	UDMA, TEGDMA
Gradia Direct flo	GC (Tokyo, Japan)	1502041	GDf	UDMA, TEGDMA
Filtek Supreme	3M ESPE (St. Paul, MN, USA)	N763255	FS	Bis GMA, TEGDMA, UDMA
Filtek Supreme flow	3M ESPE (St. Paul, MN, USA)	6033A2	Fsf	Bis GMA, TEGDMA, UDMA
TetricEvo Ceram	Ivoclar Vivadent, (Schaan, Lichtenstein)	V16037	TeC	Bis GMA, Bis EMA, UDMA, TEGDMA
TetricEvo flow	Ivoclar Vivadent, (Schaan, Lichtenstein)	V02622	Tcf	Bis GMA, Bis EMA, UDMA, TEGDMA

Bis GMA (Bysphenil-glycidyl-methacrylate), Bis EMA (Ethoxylated bisphenol A glycol dimethacrylate), UDMA (Urethane-dimethacrylate), TEGDMA (Triethylene glycoldymethacrylate), EBPADMA (ethoxylated bisphenol A dimethacrylate).

**Table 2 polymers-15-00627-t002:** Mean values (SD) of residual leachable compounds for each dental material at each time point.

Material		Compound	24 h		7 d		28 d		
	Category *		Mean	SD	Mean	SD	Mean	SD	ANOVA **
Fbf	BHV	TEGDMA	25.9	(12.9)	43.7	(28.6)	33.2	(28.3)	0.685
		Bis GMA	2.2	(0.5)	0.0	(0.0)	0.0	(0.0)	<0.001
		DMA BEE	1.0	(0.4)	2.6	(0.9)	3.5	(0.3)	0.006
		HEMA	0.0	(0.0)	0.0	(0.0)	0.0	(0.0)	NA
TecBf	BHV	TEGDMA	9.9	(13.9)	31.0	(13.7)	93.3	(66.5)	0.097
		Bis GMA	3.8	(0.9)	6.8	(0.4)	7.3	(0.5)	0.001
		DMA BEE	3.8	(0.0)	3.8	(0.0)	3.8	(0.0)	0.124
		HEMA	0.0	(0.0)	0.0	(0.0)	0.0	(0.0)	NA
Xf	BHV	TEGDMA	127.4	(98.2)	251.4	(33.3)	121.7	(63.1)	0.109
		Bis GMA	5.0	(0.6)	6.8	(0.5)	5.7	(1.1)	0.082
		DMA BEE	3.8	(0.0)	3.8	(0.0)	3.8	(0.0)	0.159
		HEMA	0.0	(0.0)	0.0	(0.0)	0.0	(0.0)	NA
Fbff	BLV	TEGDMA	2.2	(3.4)	14.8	(13.4)	1.4	(2.4)	0.154
		Bis GMA	0.0	(0.0)	4.1	(0.7)	6.9	(2.1)	0.002
		DMA-BEE	1.5	(0.9)	2.7	(0.5)	4.6	(1.0)	0.009
		HEMA	0.0	(0.0)	0.0	(0.0)	0.0	(0.0)	NA
TefBf	BLV	TEGDMA	43.6	(41.1)	22.9	(23.5)	46.6	(25.3)	0.619
		Bis GMA	7.9	(1.1)	12.1	(4.3)	14.5	(11.5)	0.552
		DMA BEE	1.9	(0.3)	3.2	(1.2)	3.8	(3.2)	0.533
		HEMA	0.0	(0.0)	0.0	(0.0)	0.0	(0.0)	NA
SDR	BLV	TEGDMA	61.0	(3.7)	54.2	(16.2)	54.2	(15.3)	0.772
		Bis GMA	0.0	(0.0)	0.0	(0.0)	0.0	(0.0)	NA
		DMA BEE	0.8	(0.4)	1.9	(0.2)	2.0	(0.2)	0.004
		HEMA	0.0	(0.0)	0.0	(0.0)	0.0	(0.0)	NA
Fs	CHV	TEGDMA	191.1	(86.6)	239.6	(27.7)	128.3	(34.3)	0.127
		Bis GMA	3.1	(2.2)	2.1	(1.2)	6.5	(6.3)	0.403
		DMA BEE	3.8	(0.0)	3.8	(0.0)	3.8	(0.0)	0.234
		HEMA	0.0	(0.0)	0.0	(0.0)	0.0	(0.0)	NA
TeC	CHV	TEGDMA	65.9	(27.8)	29.7	(38.8)	4.5	(7.7)	0.091
		Bis GMA	7.9	(0.7)	13.4	(1.1)	16.1	(1.8)	0.001
		DMA BEE	1.3	(0.2)	2.5	(0.2)	3.4	(0.5)	0.001
		HEMA	0.0	(0.0)	0.0	(0.0)	0.0	(0.0)	NA
G	CSHV	TEGDMA	55.3	(48.9)	109.2	(24.1)	98.9	(35.9)	0.255
		Bis GMA	0.0	(0.0)	0.0	(0.0)	0.0	(0.0)	NA
		DMA BEE	0.0	(0.0)	0.0	(0.0)	0.0	(0.0)	0.43
		HEMA	14.1	(1.6)	25.1	(4.0)	34.2	(10.0)	0.022
Fsf	CLV	TEGDMA	93.7	(99.3)	64.9	(4.0)	51.2	(6.0)	0.671
		Bis GMA	6.6	(1.5)	4.2	(0.5)	4.0	(0.7)	0.033
		DMA BEE	3.2	(0.7)	2.7	(0.6)	3.1	(0.3)	0.558
		HEMA	0.0	(0.0)	0.0	(0.0)	0.0	(0.0)	NA
Tcf	CLV	TEGDMA	59.2	(26.6)	175.6	(114.9)	202.8	(99.2)	0.192
		Bis GMA	17.1	(2.7)	14.4	(2.7)	15.8	(1.6)	0.428
		DMA BEE	4.1	(0.5)	4.5	(0.9)	5.0	(0.5)	0.343
		HEMA	0.0	(0.0)	0.0	(0.0)	0.0	(0.0)	NA
GDf	CLV	TEGDMA	131.8	(44.1)	85.6	(46.5)	75.1	(12.2)	0.227
		Bis GMA	0.0	(0.0)	0.0	(0.0)	0.0	(0.0)	NA
		DMA BEE	0.0	(0.0)	0.0	(0.0)	0.0	(0.0)	0.245
		HEMA	9.7	(1.1)	8.0	(0.4)	7.7	(0.5)	0.036

* BHV—Bulk high viscosity material, BLV—Bulk low viscosity, CHV—Conventional high viscosity, CLV—Conventional low viscosity. ** Kruskal-Wallis test one way analysis of variance by ranks test result *p*-value. Significant results highlighted in bold. NA—Not applicable. TEGDMA (Triethylene glycoldymethacrylate), Bis GMA (Bysphenil-glycidyl-methacrylate), DMA BEE (4-dimethylaminobenzoic acid ethyl ester), HEMA (2-Hydroxyethyl methacrylate).

**Table 3 polymers-15-00627-t003:** Concentration of different compounds leached from different types of dental material preparations from the same manufacturer after 24 h.

Manufacturer and Analyte	BHV		BLV		CHV		CLV		*t*-Test *p* Value			
	Mean	(SD)	Mean	(SD)	Mean	(SD)	Mean	(SD)	BHV vs. CHV	BLV vs. CLV	BHV vs. BLV	CHV vs. CLV
**Filtek**												
DMA BEE	1.0	(0.4)	1.5	(0.9)	3.8	(0.0)	3.2	(0.7)	0.006	0.056	0.449	0.294
Bis GMA	2.2	(0.5)	0.0	(0.0)	3.1	(2.2)	6.6	(1.5)	0.518	NA	NA	0.083
TEGDMA	25.9	(12.9)	2.2	(3.4)	191.1	(86.6)	93.7	(99.3)	0.082	0.252	0.037	0.269
HEMA	0.0	(0.0)	0.0	(0.0)	0.0	(0.0)	0.0	(0.0)	NA	NA	NA	NA
**Tetric**												
DMA BEE	3.8	(0.0)	1.9	(0.3)	1.3	(0.2)	4.1	(0.5)	0.001	0.003	0.008	0.001
Bis GMA	3.8	(0.9)	7.9	(1.1)	7.9	(0.7)	17.1	(2.7)	0.003	0.005	0.007	0.004
TEGDMA	9.9	(13.9)	43.6	(41.1)	65.9	(27.8)	59.2	(26.6)	0.036	0.611	0.249	0.779
HEMA	0.0	(0.0)	0.0	(0.0)	0.0	(0.0)	0.0	(0.0)	NA	NA	NA	NA
**Other**												
DMA BEE	3.8	(0.0)	0.8	(0.4)	0.0	(0.0)	0.0	(0.0)	NA	NA	0.006	NA
Bis GMA	5.0	(0.6)	0.0	(0.0)	0.0	(0.0)	0.0	(0.0)	NA	NA	NA	NA
TEGDMA	127.4	(98.2)	61.0	(3.7)	55.3	(48.9)	131.8	(44.1)	0.318	0.109	0.362	0.114
HEMA	0.0	(0.0)	0.0	(0.0)	14.1	(1.6)	9.7	(1.1)	NA	NA	NA	0.018

BHV—Bulk high viscosity material, BLV—Bulk low viscosity, CHV—Conventional high viscosity, CLV—Conventional low viscosity. Significant results highlighted in bold. NA—Not applicable. TEGDMA (Triethylene glycoldymethacrylate), Bis GMA (Bysphenil-glycidyl-methacrylate), DMA BEE (4-dimethylaminobenzoic acid ethyl ester), HEMA (2-Hydroxyethyl methacrylate).

**Table 4 polymers-15-00627-t004:** Concentration of different compounds leached from different types of dental material preparations from the same manufacturer after 28 days.

Manufacturer and Analyte	BHV		BLV		CHV		CLV		*t*-Test *p* Value			
	Mean	(SD)	Mean	(SD)	Mean	(SD)	Mean	(SD)	BHV vs. CHV	BLV vs. CLV	BHV vs. BLV	CHV vs. CLV
**Filtek**												
DMA BEE	3.5	(0.3)	4.6	(1.0)	3.8	(0.0)	3.1	(0.3)	0.295	0.067	0.156	0.049
Bis GMA	0.0	(0.0)	6.9	(2.1)	6.5	(6.3)	4.0	(0.7)	NA	0.084	NA	0.567
TEGDMA	33.2	(28.3)	1.4	(2.4)	128.3	(34.3)	51.2	(6.0)	0.021	<0.001	0.192	0.019
HEMA	0.0	(0.0)	0.0	(0.0)	0.0	(0.0)	0.0	(0.0)	NA	NA	NA	NA
**Tetric**												
DMA BEE	3.8	(0.0)	3.8	(3.2)	3.4	(0.5)	5.0	(0.5)	0.323	0.547	0.991	0.022
Bis GMA	7.3	(0.5)	14.5	(11.5)	16.1	(1.8)	15.8	(1.6)	0.001	0.865	0.389	0.829
TEGDMA	93.3	(66.5)	46.6	(25.3)	4.5	(7.7)	202.8	(99.2)	0.149	0.057	0.319	0.075
HEMA	0.0	(0.0)	0.0	(0.0)	0.0	(0.0)	0.0	(0.0)	NA	NA	NA	NA
**Other**												
DMA BEE	3.8	(0.0)	2.0	(0.2)	0.0	(0.0)	0.0	(0.0)	NA	NA	0.003	NA
Bis GMA	5.7	(1.1)	0.0	(0.0)	0.0	(0.0)	0.0	(0.0)	NA	NA	NA	NA
TEGDMA	121.7	(63.1)	54.2	(15.3)	98.9	(35.9)	75.1	(12.2)	0.615	0.137	0.146	0.339
HEMA	0.0	(0.0)	0.0	(0.0)	34.2	(10.0)	7.7	(0.5)	NA	NA	NA	0.045

BHV—Bulk high viscosity material, BLV—Bulk low viscosity, CHV—Conventional high viscosity, CLV—Conventional low viscosity. Significant results highlighted in bold. NA—Not applicable. TEGDMA (Triethylene glycoldymethacrylate), Bis GMA (Bysphenil-glycidyl-methacrylate), DMA BEE (4-dimethylaminobenzoic acid ethyl ester), HEMA (2-Hydroxyethyl methacrylate).

## Data Availability

The datasets generated and analyzed during the current study are available from the corresponding author on reasonable request.

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
