# Peer review of "Detection of Leachable Components from Conventional and Dental Bulk-Fill Resin Composites (High and Low Viscosity) Using Liquid Chromatography-Tandem Mass Spectrometry (LC-MS/MS) Method"

_polymers, 2023, doi:10.3390/polym15030627_

Round 1

Reviewer 1 Report

Very interesting research, the subject of the safety of composite materials is very current and important. Happy to read this as a reviewer. A few remarks below

Abstract

HEMA-  please use 2 hydroxy ethyl methacrylate

TEGDMA, DMA-29 BEE and Bis-GMA- please use the full names.

Bis-GMA was found in Filtek Bulk Fill, Filtek Bulk Fill 25 flow, X tra fill, Filtek Supreme, Filtek Supreme flow, Tetric EvoCeram Bulk fill, Tetric EvoFlow Bulk 26 fill, TetricEvo flow and TetricEvo Ceram samples- Maybe it's better to write it a little differently. Bisphenol A-glycidyl methacrylate is found in Filtek and Teric materials

I would also add a conclusion : Very important is the time of polymerization and the thickness of the layer for the amount of unpolymerized monomer. Clinical conclusions are important for dentists.

Introduction.

Line 50-55

For the sake of order, it would be good to give the full names of the materials, e.g. UDMA - Urethane dimethacrylate etc. TPO, PPD …. DEGMA….

A similar problem also occurs in methacrylate resins used in 3D printing. Which is also worth mentioning? This is a very fashionable topic right now. Good parts of compounds can also be used UV Vis, although your method is better

González, G.; Barualdi, D.; Martinengo, C.; Angelini, A.; Chiappone, A.; Roppolo, I.; Pirri, C.F.; Frascella, F. Materials Testing for the Development of Biocompatible Devices through Vat-Polymerization 3D Printing. Nanomaterials 2020, 10, 1788

 Raszewski, Z.; Kulbacka, J.;Nowakowska-Toporowska, A.Mechanical Properties, Cytotoxicity, and Fluoride Ion Release Capacity of Bioactive Glass-Modified Methacrylate Resin Used in Three-Dimensional Printing Technology. Materials 2022, 15, 1133. https://doi.org/10.3390/ma15031133

Materials and methods

it would be clearer to present the materials used in a table and add the  info from SDS about composition.  It would be a kind of confirmation that what you have given is declared by the producers, at least partially

Name

producer

Lot

aberration

Composition declared by producer

Tetric Evo Ceram Bulk Fill

Ivoclar Vivadent, (Schaan Lichtenstein)

LOT 82 O135539/

TeCBf

Was the lamp power measured with a radiometer? this will affect the degree of cross-linking and thus the amount of leached monomers

Was the alcohol solution changed or was it the same 28 days, if so, the values of the eluted components will be different. We have saliva in our mouth to dilute it.

Results.

Gradia direct flo- Gradia Direct Flo

Line 138

LOQ- bellow detection?

Line 148

Gradia (from 14.1+1.6 to  34.2+10; ANOVA p=0.022- what units??

Line 175

BiS-GMA- bis GMA

Table 1, Table 2, Table 3 please specify units, whether they were micrograms or even smaller amounts or %

Discussion

Guertsen or Gutrsen because in 24 is the Gutrsen?

Reference

It is also good to consult the literature from recent years

Janani, K.; Teja, K.V.; Sandhya, R.; Alam, M.K.; Al-Qaisi, R.K.; Shrivastava, D.; Alnusayri, M.O.; Alkhalaf, Z.A.; Sghaireen, M.G.; Srivastava, K.C. Monomer Elution from Three Resin Composites at Two Different Time Interval Using High Performance Liquid Chromatography—An In-Vitro Study. Polymers 2021, 13, 4395

Nazar AM, George L, Mathew J. Effect of layer thickness on the elution of monomers from two high viscosity bulk-fill composites: A high-performance liquid chromatography analysis. J Conserv Dent 2020;23:497-504

good luck with your further research, I'm looking forward to your next research

Author Response

Please find submitted the revised manuscript " DETECTION OF LEACHABLE COMPONENTS FROM CONVENTIONAL AND DENTAL BULK-FILL RESIN COMPOSITES (HIGH AND LOW VISCOSITY) USING LIQUID CHROMATOGRAPHY-TANDEM MASS SPECTROMETRY (LC-MS/MS) METHOD".

We appreciate considering our manuscript for publication in Polymers. On behalf of the coauthors and myself, I want to thank the reviewers for constructive criticism of the manuscript, which certainly contributed to the quality of the manuscript. We have thoroughly considered all the remarks given and we find them constructive and helpful in improving the manuscript. The manuscript has been corrected accordingly, as it follows in Word document.

Reviewer 2 Report

I have reviewed the manuscript, "DETECTION OF LEACHABLE COMPONENTS FROM CONVENTIONAL AND BULK FILL COMPOSITES (HIGH AND LOW VISCOSITY) USING LIQUID CHROMATOG-RAPHY-TANDEM MASS SPECTROMETRY (LC-MS/MS ) METHOD" for its contents. Authors have investigated the investigate leachable components (monomers) in high and low-viscosity bulk-fill and conventional resin composite materials after polymerization. 

The introduction does not provide any information to support the objective of study.

Unfortunately, I don’t find any novel information in the manuscript. The data present is insufficient for a full-length research paper. 

You can find the literature is exhaustive with similar studies. 

https://scholar.google.com/scholar?q=LEACH+FROM+CONVENTIONAL+AND+BULK+FILL+COMPOSITES&hl=en&as_sdt=0&as_vis=1&oi=scholart

https://onlinelibrary.wiley.com/doi/abs/10.1034/j.1600-0722.2000.108004341.x

Even, a recent systematic review on this issue has been published:

https://www.ncbi.nlm.nih.gov/pmc/articles/PMC9525379/

https://onlinelibrary.wiley.com/doi/abs/10.1002/jbm.b.34843

In my opinion, the research question addressed by the authors is well discussed by previously published studies. therefore, it does not address a specific gap in the field, and not going to add any new information to the existing science on this topic. 

Author Response

(The authors gave the same response as above.)

Reviewer 3 Report

Manuscript ID: polymers-2059151

Detection of leachable components from conventional and bulk fill composites (high and low viscosity) using liquid chromatography tandem mass spectrometry (LC-MS/MS) method

Main concerns: What are the novelties of this work? What was the benefit of the usage of tandem mass spectrometry as an additional method besides HPLC? Although the authors emphasize the method used, however its benefit does not appear in the communication. The basis of comparison between materials (grouping according to manufacturers) is not appropriate. Intensive English correction is necessary.

Abstract

-Monomer elution from resin composites is an evidence based statement. What is the additional information to the already known facts?

-According to the literature and the results of this study, monomers are eluted from all resin composite material. How can a dentist properly select material knowing the above fact? The authors' advice as a conclusion is meaningless.

-Correct name of composite (too general) is: resin composite or resin-based composite or dental composite

Introduction

-Please correct the chemical name of monomers and at the first mentioning please give the full chemical name of all the monomers, initiators and additional components before giving the abbreviation.

-Not only few, but several publications deal with the monomer elution from both conventional and bulk-fill resin composites. Please, introduce more carefully and comprehensively the main objective of the current study. What we know so far thanks to the results of the investigations and what is incomplete and requires further investigations?

-Please also formulate a hypothesis for the possible differences between the time intervals.

Materials and Methods

-Please, provide a table with the compositions of the investigated resin composites.

-All the samples have been made in 2 mm thickness? As a control, you can use bulk-fills in 2 mm, however, if you want to compare the real monomer elution from conventional and bulk-fills (as it can represent a clinical situation) bulk must be investigated in 4 mm as well.

-Detailed description of the method is necessary. How have you determined quantitatively the monomer elution? Please describe the validation and calibration.

-Why only TEGDMA, HEMA, CQ and DMABEE were determined quantitatively? Why the other detected components were not presented and measured?

-Please, provide a sample size calculation.

Results

-The description of the results is confused and hard to clearly understand. It is recommended to make bar diagrams of the categorized materials by comparing the elution at different time points.

-Tables 1: A table should be self-explanatory, thus, please give the full name of all the abbreviations.

-Table 2, 3, Figure 1: There is no point in giving a summarized elution value per manufacturer, since it is not really a manufacturer-dependent phenomenon (only to a very small extent), but depends on the individual composition and elution circumstances. It is much more fortunate to compare bulk materials with traditional, high viscosity to the flowable, etc. 

Discussion

-The discussion section is quite short, does not sufficiently discuss the questions that arise, and compares the results with very few (some of them old) similar studies. The authors refer to the fact that there is only one research in the literature that also uses liquid chromatography tandem mass spectrometry, even though with a careful search many researches are found using this technique.

Conclusions

-The conclusions have been drawn are not correct and do not help the practitioners in material selection.

Author Response

(The authors gave the same response as above.)

Reviewer 4 Report

December, 20, 2022

Dear authors

Thank you for an interesting report.

In this study, you examined leachable components (monomers) in high and low viscosity bulk-fill and conventional resin composite materials after polymerization by using each 6 materials.

I agree to many parts of your claims and guessed that the subject of this paper will be of interest to the readership of Polymers. However, I think that several revisions are required as follows:

1. Introduction

Various advantages of bulk-filled composite materials are described, but the differences from the conventional type in terms of components, the reason why bulk filling is possible, etc. are not explained. I think you had better add this point.

2. Materials and Methods

1. To make it easier for the reader to read this article, I think you should provide a table showing the composite materials used.

2. Similar to 1, I think you should insert an image of the prepared specimen with dimensions, and/or an image of the specimen under immersion, in order to make it easier for the reader to understand this article. As it is now, you only express them in sentences, so it is difficult for readers to imagine specimens and immersion tests.

3. Results

Regarding the values shown in Table 1, I think that you can suggest that translating these values into a value of leachable component value versus immersion time curve for each product would give the readers better understanding of the timing of each component’s exudation within the product and between products. I think that they can inform the readers at a glance. And, if possible, I would like the you to attach the data in Table 1 to the article as supplementary material.

4. Discussion

You explain that the 75% ethanol solution used as the immersion solution was selected because of the similarity of the solubility parameters and the fact that it is also used by other researchers. However, this solution is clearly different from saliva components (inorganic components such as calcium, phosphate, sodium, etc., and organic components such as mucin, antibacterial and immunological substances, etc.). Is it possible to consider the leach behavior of these materials in the oral cavity in the same way as this result? I think that factors other than solubility are intricately intertwined. I think your thoughts should be added on that point.

Author Response

(The authors gave the same response as above.)

Reviewer 5 Report

This paper reports LC-MS/MS to detect leached monomers from a set of polymerized commercial dental composites. The main concern with this study is the lack of novelty surrounding it. There are already many different studies, published in the literature (and in very good journals), where this characterization has been made. The conclusions of this study are not in any way novel or interesting. The authors conclude that the monomers present in the chemical composition of the bulk-fill composites may be eluted after polymerization. It is safe to say that all researchers already know this. Furthermore, the authors do not make mention of important studies that have tested this already.

Please see:

https://www.sciencedirect.com/science/article/pii/S0109564115001177?casa_token=lxGdiDf-FeYAAAAA:x7rXrgJjrHQAbV9ercMt6yStmwZViCZcg6f-EfmZJh12OLS8eYZYbByihRC3qfW5z5nBKlQRrWc

https://www.sciencedirect.com/science/article/pii/S0109564115004364?casa_token=LpI9G4yngGUAAAAA:YSw6RFixKYVxkkTRQf0YLCiabIFgwNLteMhNrHZtmpZlA5Clo7KP5I7_KSkmBIxVRsOEEr_ecSM

https://www.jstage.jst.go.jp/article/josnusd/62/3/62_19-0221/_article/-char/ja/

And many other examples. I do not recommend this paper to be published, since it does not report novel research.

Author Response

(The authors gave the same response as above.)

Reviewer 6 Report

This is an interesting study determining the amount of monomer release from 12 composite resin products. The aim was clearly defined. The study design and the methodology matched the objective of the study. Here are some questions about the study.

How do you define bulk fill and conventional composite?

Please provide the significance of the study by discussing the consequence of different types of monomer leakage.

Please provide the component of products and their proportion to provide the reference for the results.

How did you determine the time point to measure the leached monomer? Why not measure for longer?

Why the amount of monomer release was lower in 28 d result compared to 7 d result or even 1 d result in some measurements in Table 1?

The references could be more updated.

Author Response

(The authors gave the same response as above.)

Round 2

Reviewer 2 Report

Dear Authors,

Many thanks for the revision, the introduction and the discussion sections are improved. However, the question of this research's novelty and impact remains. As mentioned, quadrupole tandem mass spectrometry (LC-MS/MS) method was used however, comparing finding with other/similar techniques is still missing. 

Reviewer 3 Report

The manuscript has been improved for publication, but there is one more comment: If the authors did not compare different layer thicknesses and different exposure times they cannot state a conclusion about these parameters. 

Reviewer 5 Report

Even though the authors did make some changes in the manuscript, its novelty is still questionable, and the main flaw is related to the research design. I still believe this is not worthy of a full-text paper in Polymers.

Round 3

Reviewer 2 Report

thank you for the response